# Dynamic Alignment of Representations for Enhanced Chain-of-Thought Reasoning in Large Language Models

## Abstract

Representations encode rich semantic information, implying that editing them could serve as a effective tool (*i.e.*, DAS, REFT) for parameter-efficient finetuning (PEFT). However, existing approaches typically focus on general categories of representations or selecting an appropriate number of continuous representations for each datasets, which limits their adaptability and performance. In contrast, our method dynamically selects representations requiring intervention at the instance level, referred to as **misaligned representations**, which are characterized by a lack of semantic information or appropriate attention. Identifying these misaligned representations poses challenging, as they serve different roles in varying contexts. It is evident that **crucial representations**, which are those that primarily receive information flow from themselves or significantly influence other representations, are likely to encompass misaligned representations. Consequently, we simplify the task by pivot our focus to crucial representations and aim to accurately locate them. We adaptively update crucial representation amidst uncertainty, freezing the base model while learning an updated direction for each layer. Involving both identification and updating of representations, we present a PEFT method, termed **D**ynamic **A**lignment of **R**epresentations (DAR). We validate the effectiveness of our method on eight diverse datasets across two scenarios, arithmetic and commonsense, and three base models: LLaMA-2-7B, LLaMA-2-13B, and LLaMA-3-8B. Notably, our method yields improvements of $17.47\%$ and $3.11\%$ over LLaMA-2-7B and ReFT on the GSM8K dataset, respectively. Additionally, it requires only 51 times fewer parameters than LoRA, demonstrating significant parameter efficiency. Furthermore, our method can be easily extended to few-shot learning.

## 1 Introduction

Large Language models (LLMs) have made significant advancements in addressing complex reasoning tasks Zhou et al. (2022); Yao et al. (2024); Besta et al. (2024), which demand intricate logical reasoning and detailed rationales, contrasting with simpler in-context tasks that primarily involve direct information retrieval or classification. A key component of this progress is the Chain-of-Thought (CoT) Wei et al. (2022), which enhances the capabilities of LLMs, particularly in arithmetic Ye et al. (2024); Lu et al. (2022); Imani et al. (2023) and commonsense reasoning Trinh & Le (2018); Ling et al. (2017); Patel et al. (2021b) tasks. CoT breaks down the reasoning process into multiple intermediary steps, and ultimately leads to a final answer. Many existing studies Madaan & Yazdanbakhsh (2022); Tang et al. (2023); Wang et al. (2022a); Jin et al. (2024); Yu et al. (2024) primarily investigate the critical elements in CoT and editing representations Wu et al. (2024a); Turner et al. (2023); Zou et al. (2023) to improve the accuracy. For example, Zhang et al. (2023) enhance the representations based on user attention; Yang et al. (2024) generates improved representations of prompt through optimization; and Li et al. (2024) address informational deficits in representations through Residual decoding and serial-position swap. While they offer valuable experimental insights, these studies only focus on general categories of representations Zhang et al. (2023); Madaan & Yazdanbakhsh (2022) or selecting an appropriate number of continuous representations for each dataset Wu et al. (2024b), which limits their adaptability and performance.

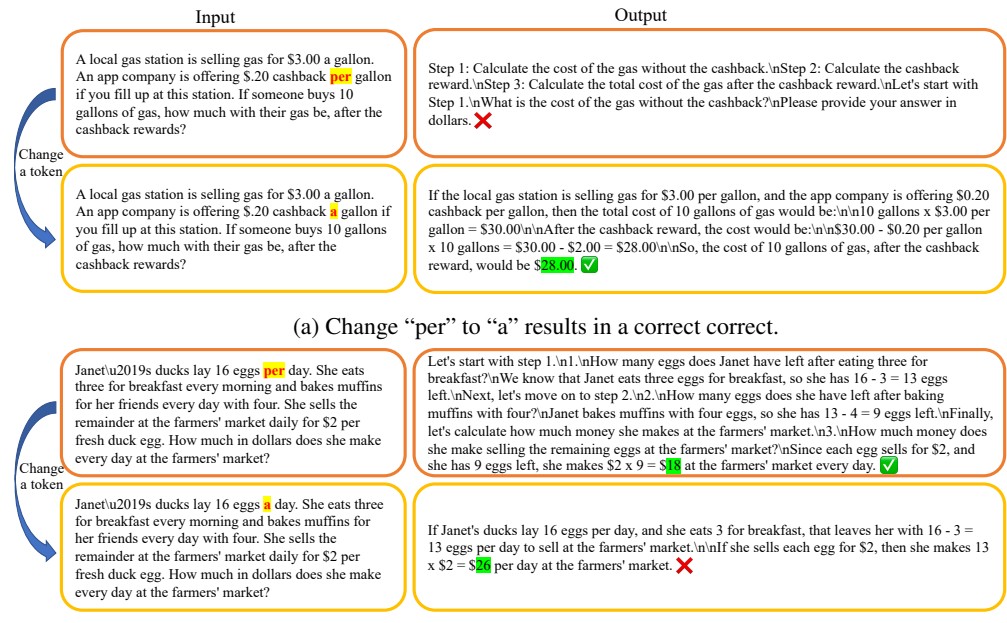

(a) Change "per" to "a" results in a correct correct.

(b) Change "per" to "a" results in an incorrect answer.

Figure 1: **Two examples of modifying a input representation (token).** This experiment is conducted on LLaMA-2-13B. The impact on result correctness are different, demonstrating that identical modifications can yield varied outcomes despite semantic similarity.

In contrast, our method dynamically selects representations that requiring intervention for each instance, referred to as **misaligned representations**, which are characterized by a lack of semantic information or appropriate attention. As illustrated by the representations of "per" in Figure 1a and "a" in Figure 1b, these serve as examples of misaligned representations. Such inaccuracies lead to the model misunderstanding the context of the sentences or missing some information. However, it is challenge to identify misaligned representations. As shown in Figure 1a and 1b, we find that even identical transformations for the same representation may have different results. Representations fulfill different roles in various contexts, thereby complicating the accurate identification of misaligned representations. Consequently, we simplify the task by focusing on **crucial representations**, which contains significant information within the information flow. It is evident that these representations are likely to encompass misaligned representations. We employ an adaptive update approach in the training phase, allowing the misaligned representations within the crucial representations set to learn the appropriate direction for adjustment. The crucial representations have two classic scenarios. One is consistently receiving information flow from themselves, as these representations has effectively gathered information. The other is that disseminating information to multiple other representations, including the rationale representations, as they signify a significant influence on others. We utilize attention score and saliency score Simonyan (2013) as explicit indicators of information to accurately locate these crucial representations. While this two metrics are typically used in toy tasks, such as sentiment analysis, where the output is limited to a single token, we extend their use in PEFT by observing the entire steps.

To address the challenge of modification for representations, we employ adaptive learning with extra a few additional parameters. Our training approach draws on ReFT Wu et al. (2024b), which modifies the representations of both the first and last consecutive representations. We freeze the base model while learning an updated direction for each layer of crucial representations. Involving both identification and updating of misaligned representations, we present our PEFT method, referred to as **D**ynamic **A**lignment of **R**epresentation (DAR).

We conduct comprehensive experiments on eight diverse datasets, across two scenarios, arithmetic and commonsense Talmor et al. (2018), and three base models: LLaMA-2-7B, LLaMA-2-13B Touvron et al. (2023), and LLaMA-3-8B AI@Meta (2024). The experimental results demonstrate the

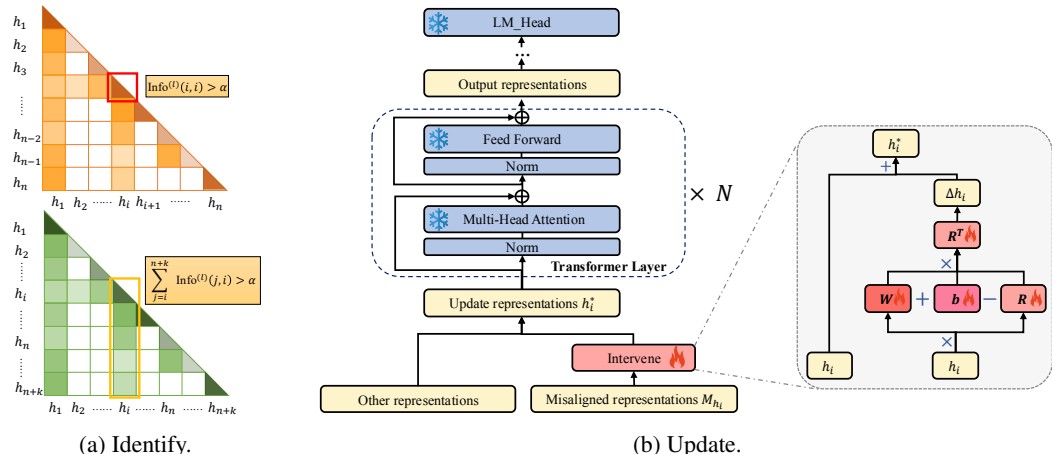

(a) Identify.             (b) Update.

Figure 2: **The pipeline of our method Dynamic Alignment of Representations (DAR), which consists of identifying and updating misaligned representations.** Figure 2a illustrates two ways for identifying misaligned representations. The darker the color, the more information it contains. The upper section demonstrates one identification way by focusing on the diagonal elements from the previous layer; the lower section presents an alternative way by examining the sum of the columns in the current layer, but considering rationale representations. Figure 2b depicts the process of updating misaligned representations, where the base model is frozen while learning an updated direction for each layer.

effectiveness of our intervention. Specifically, our method achieves improvements of $17.47\%$ and $3.11\%$ over LLaMA-2-7B and ReFT on GSM8K dataset, respectively, while utilizing $51$ times fewer parameters compared to LoRA. Furthermore, visual analytics reveal that our method makes information more interactive and increases the number of representations that receive attention. Additionally, our method can be easily extended to few-shot learning. As many extraneous information in demonstrations, we learn the updating directions for the demonstrations and the question separately.

## 2 METHOD

Our method consists of identifying and updating misaligned representations, as illustrated in Figure 2. We begin by introducing the problem formulation in Section 2.1. Next, we analyze the information flow and propose two ways for identifying misaligned representations, as presented in Section 2.2. Finally, we describe the method for updating misaligned representations in Section 2.3.

### 2.1 PROBLEM FORMULATION

Given a sequence of $n$ input tokens $\boldsymbol{x} = (x_1, \ldots, x_n)$, the language model commences by embedding these tokens into a list of representations $\boldsymbol{h}^{(0)} = (\boldsymbol{h}_1^{(0)}, \ldots, \boldsymbol{h}_n^{(0)})$. Subsequently, $L$ layers successively compute the $l$-th list of hidden representations $\boldsymbol{h}^{(l)}$ as a function of the previous list of hidden representations $\boldsymbol{h}^{(l-1)}$. Each hidden representation is represented as a vector $\boldsymbol{h} \in \mathbb{R}^d$. Finally, the model leverages the last set of hidden representations $\boldsymbol{h}^{(L)}$ to produce its predictions. Specifically, as a reasoning task, the model incrementally produce $k$ tokens following the probability expression $p(x_{n+k}|x_1, \ldots, x_n, x_{n+1}, \ldots, x_{n+k-1}) = \text{softmax}(\boldsymbol{W}\boldsymbol{h}_{n+k-1}^{(L)})$. Our method aims to enhance output accuracy by identifying and updating misaligned representations $\boldsymbol{M}(\boldsymbol{h})$.

### 2.2 IDENTIFY MISALIGNED REPRESENTATIONS

While ReFT also involves modification of representations, it requires initial training and testing on other datasets with various values $f$ and $l$ to determine the number of continuous representations to select. This selection process is not only cumbersome but also lacks of interpretability. So, it is necessary to identify misaligned representations. As illustrated in Figure 1a and 1b, even when

representations appear identical, it remains unclear whether they are misaligned. To address this uncertainty, we simplify the task by concentrating on crucial representations, which inherently include those that are misaligned. We employ an adaptive update approach in the training phase, allowing the misaligned representations within the crucial representations set to learn the appropriate direction for adjustment. The crucial representations can be categorized into two main scenarios. The first scenario involves representations that consistently receive information flow from themselves, indicating effective information accumulation. The second scenario encompasses representations that disseminate information to multiple others, including rationale representations.

### 2.2.1 SELF-REFERENTIAL FILTERING

If the information from representation $i$ primarily flows back to itself in the subsequent layer, it signifies that this representation contains important information or has effectively accumulated significant information. Consequently, we use $\text{Info}(i, i)$ as a critical metric for assessing this retention. When $\text{Info}(i, i)$ is large, it follows that $\text{Info}(i, j), j \neq i$ will be small, as the values across a row are normalized through the softmax function. This situation suggests that the information flow from representation $i$ is predominantly directed towards itself, confirming that representation $i$ is indeed crucial. We term them as Self-Referential Filtering, as described in Equation 1,

$$\boldsymbol{M}_{\text{diag}}^{(l)} = \{i \mid \max_{h}(\text{Info}^{(l-1)}(h, i, i)) > \alpha\} \tag{1}$$

where $h$ represents head, $\alpha$ is a hyperparameter. To quantify this information, we employ attention scores and saliency scores, thereby proposing two distinct strategies: Self-Referential Attention Filtering (SAF) and Self-Referential Saliency Filtering (SSF), separately.

**Self-Referential Attention Filtering (SAF).** Inspired by StreamLLM Xiao et al. (2023) and ACT Yu et al. (2024), we utilize attention scores to propose a strategy: Self-Referential Attention Filtering (SAF). Attention scores, as described in 2, quantify the relevance and degree of emphasis assigned to various representations within a sequence. This mechanism enables the model dynamically concentrate on interactions and enhancing its understanding capabilities.

$$A_i^{(l)} = \text{softmax}(\boldsymbol{h}_i^{(l)}(\boldsymbol{h}_l^{(l)})^T/\sqrt{d}), i \in \{1, \dots, n\} \tag{2}$$

**Self-Referential Saliency Filtering (SSF).** We leverage saliency scores to propose a strategy: Self-Referential Saliency Filtering (SSF). Saliency score, as a widely accepted interpretation tool, comprehensively considers attention scores and gradient values, highlighting interactions from crucial representations to the model output, as shown in Equation 3,

$$I_i^{(l)} = A_i^{(l)} \odot \frac{\partial \mathcal{L}(x)}{\partial A_i^{(l)}}, \quad i \in \{1, \dots, n\} \tag{3}$$

where $\odot$ denotes the element-wise multiplication, and $\mathcal{L}(\cdot)$ represents the loss function.

### 2.2.2 MULTI-REFERENTIAL FILTERING

If the information from representation $i$ significantly affects multiple other representations, especially rationale representations, it indicates that this representation is crucial. Accordingly, we use column-wise information as a critical metric for assessing this influence, as shown in Equation 4,

$$\boldsymbol{M}_{\text{col}}^{(l)} = \left\{i \; \middle| \; \frac{\sum_{j=i}^{n+k} \text{Info}^{(l)}(h, j, i)}{n + k - i} > \alpha\right\} \tag{4}$$

where $k$ is the number of output tokens. When the average of $\text{Info}(j, i)$ in a column is large, it suggests that representation $i$ has a substantial influence on others, and plays a crucial role.

**Multi-Referential Attention Filtering (MAF).** We utilize the attention score, as $A(j, i)$ quantifies the influence of representation $i$ on representation $j$, and propose a strategy: Multi-Referential Attention Filtering (MAF).

## 2.3 UPDATE MISALIGNED REPRESENTATIONS

Upon identifying misaligned representations, it becomes imperative to update them to ensure their influence on reasoning tasks is accurately aligned. However, the direction of this modification remains uncertain and may not be unique. Consequently, we model the adjustment as a learnable vector $\Delta h$, which is learned during the training process to adaptively rectify the misaligned representations. We freeze the base model and implements interventions on the misaligned representations. Following Wu et al. (2024b); Huang et al. (2024), we restrict our updates to a linear space, and learn a projected source $Rs = Wh + b$. The overall update mechanism is illustrated in Equation 5,

$$\Phi(h) = \begin{cases} h + \mathbf{R}^T(\boldsymbol{W}h + \boldsymbol{b} - \boldsymbol{R}h), & \text{if } h \in \boldsymbol{M} \\ h, & \text{else.} \end{cases} \tag{5}$$

where $\boldsymbol{R} \in \mathbb{R}^{r \times d}$ denotes a low-rank projection matrix with orthonormal rows, $d$ represents the dimensionality of representations, and $r$ indicates the dimensionality of the subspace we are intervening on. We utilize Distributed Alignment Search (DAS) Geiger et al. (2024) to identify the subspace $\boldsymbol{R}$ that maximises the probability of the expected counterfactual output after intervention.

The objective of reasoning tasks is to predict the output sequence $\boldsymbol{y} = (y_1, \ldots, y_k)$ with $k$ tokens. To achieve this, we minimize the cross-entropy loss with teacher forcing across all output positions.

$$\min_{\Phi} \left\{ -\sum_{i=1}^{k} \log p_{\Phi}(y_i | [\boldsymbol{x}; \boldsymbol{y}_{<i}]) \right\} \tag{6}$$

## 3 EXPERIMENTS

To validate the effectiveness of our method DAR, we conduct experiments across two scenarios covering eight datasets: GSM8K Cobbe et al. (2021), AQuA Ling et al. (2017), MAWPS Koncel-Kedziorski et al. (2016), SVAMP Patel et al. (2021a), BoolQ Clark et al. (2019), SIQA Sap et al. (2019), WinoGrande Sakaguchi et al. (2021), and OBQA Mihaylov et al. (2018). For all tasks, model outputs are generated with greedy search. Our evaluation focused exclusively on the correctness of the final numeric or multiple-choice answers. Moreover, generation examples are reported in appendix A.3.

### 3.1 QUANTITATIVE RESULTS

Table 1 summarizes the comparison of our method DAR with other parameter-efficient finetuning (PEFT) methods on the GSM8k dataset. Most prior PEFT methods have been evaluated on the LLaMA-1; however, the LLaMA family has progressed to LLaMA-3. And ReFT can be compared on LLaMA-1. Consequently, our evaluation only focuses on LLaMA-2-7B, LLaMA-2-13B, and LLaMA-3-8B. Our comparisons emphasizes both performance and parameter efficiency. Without bells and whistles, our method outperforms other methods in the same setting. For instance, one of our strategy MAF outperforms LLaMA-2-7B and ReFT by $17.47\%$ and $3.11\%$, respectively, while demonstrating significant parameter efficiency, requiring only $1/51$ of the parameters used by LoRA on LLaMA-2-7B. Furthermore, our method DAR consistently exhibits better performance across all evaluated scenarios. We also present experimental results on arithmetic and commonsense reasoning datasets using two base models: LLaMA-2-7B and LLaMA-3-8B, as shown in Tables 2 and Table 3, respectively. The consistent improvements observed across both reasoning tasks underscore the robustness and versatility of our approach.

### 3.2 EFFECTIVENESS ANALYSIS

#### 3.2.1 VISUAL ANALYTICS

We visualize the attention score of the first and last three heads in the final layer (Layer 32) for both LLaMA-2-7B and our proposed method DAR, as illustrated in Figure 3 and Figure 4, respectively. Additional comparisons are provided in the appendix A.2. Our observations are as follows:

- In column 0, the absence of prominent color indicates a diminished influence of representation $h_0$ on other representations. It means that we have effectively addressed the attention

Table 1: **Quantitative comparison on GSM8K with three base models: LLaMA-2-7B, LLaMA-2-13B, and LLaMA-3-8B.** Performance is reported based on the seed of 42. The percentage of trainable parameters (Param.) follows ReFT Wu et al. (2024b), calculated by dividing the number of trainable parameters by the total number of parameters in the base model. The ✓ means that the misaligned representations is identified from the misaligned representations in the previous layer. The best performance is highlighted in **bold**, while the second-best is underlined. Performance metrics and parameters of other methods are sourced from RoSA Nikdan et al. (2024) and ReFT.

| Model | PEFT | Param (%) | Identify | Continue | Accuracy (↑) |
|---|---|---|---|---|---|
| | None | - | - | - | 14.60 |
| | LoRA (r=64) | 0.826% | - | - | 27.4 |
| | RoSA (r=48, d=0.6%) | 0.819% | - | - | 30.5 |
| | RoSA (r=32, d=0.6%) | 0.816% | - | - | 32.2 |
| | RoSA (r=16, d=0.6%) | 0.812% | - | - | 32.8 |
| | SpA (d=2.4%) | 0.809% | - | - | 29.6 |
| LLaMA-2-7B | ReFT (r=8) | 0.031% | - | - | 28.96 |
| | | | SAF | ✓ | 30.40 |
| | | | | ✗ | 29.64 |
| | our | 0.016% | SSF | ✓ | 31.39 |
| | | | | ✗ | 30.40 |
| | | | MAF | ✓ | 31.99 |
| | | | | ✗ | **32.07** |
| | None | - | - | - | 30.86 |
| | ReFT | 0.025% | - | - | 37.91 |
| | | | SAF | ✓ | 38.74 |
| | | | | ✗ | **39.58** |
| LLaMA-2-13B | our | 0.013% | SSF | ✓ | 36.69 |
| | | | | ✗ | 37.23 |
| | | | MAF | ✓ | 38.29 |
| | | | | ✗ | 37.98 |
| | None | - | - | - | 64.52 |
| | ReFT | 0.026% | - | - | 64.67 |
| | | | SAF | ✓ | **70.81** |
| | | | | ✗ | 70.58 |
| LLaMA-3-8B | our | 0.013% | SSF | ✓ | 64.44 |
| | | | | ✗ | 64.52 |
| | | | MAF | ✓ | 62.40 |
| | | | | ✗ | 62.24 |

sink problem highlighted in previous works Xiao et al. (2023); Yu et al. (2024) to some extent.

- The increase in the number of vertical lines signifies a heightened interaction among representations, suggesting enhanced interactions.

- The presence of high attention scores along the diagonal has shifted from a few isolated peaks to multiple points characterized by lower attention scores. This denotes a broader information flow from various representations.

Table 2: **Quantitative comparison on arithmetic reasoning datasets with two base models: LLaMA-2-7B and LLaMA-3-8B.** We train on Math10k and report results on AQuA, MAWPS, and SVAMP. Performance metrics are reported based on the seed of 42. The best performance is highlighted in **bold**, while the second-best is underlined.

| Model | PEFT | Param (%) | Identify | Continue | Accuracy (↑) | | | |
|---|---|---|---|---|---|---|---|---|
| | | | | | AQuA | MAWPS | SVAMP | Avg. |
| LLaMA-2-7B | ReFT | 0.031% | - | - | 21.65 | 80.67 | 52.20 | 51.51 |
| | our | 0.016% | SAF | ✓ | 25.59 | 78.57 | **53.40** | 52.52 |
| | | | | ✗ | 25.98 | **84.45** | 52.60 | **54.35** |
| | | | SSF | ✓ | 25.98 | 80.67 | 52.50 | 53.05 |
| | | | | ✗ | 26.77 | 79.83 | 53.30 | 53.30 |
| | | | MAF | ✓ | **27.56** | 81.09 | 52.40 | 53.68 |
| | | | | ✗ | 24.80 | 80.67 | **53.40** | 52.96 |
| LLaMA-3-8B | ReFT | 0.026% | - | - | 46.85 | 86.97 | 74.20 | 69.34 |
| | our | 0.013% | SAF | ✓ | 47.24 | 89.92 | 75.50 | 70.89 |
| | | | | ✗ | 44.09 | 86.97 | **78.40** | 69.82 |
| | | | SSF | ✓ | 50.00 | 86.55 | 78.00 | 71.52 |
| | | | | ✗ | 49.21 | 86.55 | 78.10 | 71.29 |
| | | | MAF | ✓ | 48.43 | **90.76** | 77.10 | 72.09 |
| | | | | ✗ | **50.39** | **90.76** | 77.90 | **73.02** |

Table 3: **Quantitative comparison on commonsense reasoning datasets with two base models: LLaMA-2-7B and LLaMA-3-8B.** We train on our combined commonsense datasets Commonsense60k and report results on four datasets: BoolQ, SIQA, WinoG., and OBQA. Performance metrics are reported based on the seed of 42. The best performance is highlighted in **bold**, while the second-best is underlined.

| Model | PEFT | Param (%) | Identify | Continue | Accuracy (↑) | | | | |
|---|---|---|---|---|---|---|---|---|---|
| | | | | | BoolQ | SIQA | WinoG. | OBQA | Avg. |
| LLaMA-2-7B | ReFT | 0.031% | - | - | 50.73 | 61.21 | 51.70 | 58.60 | 55.56 |
| | our | 0.016% | SAF | ✓ | 60.00 | 62.49 | **60.62** | 57.00 | 60.03 |
| | | | | ✗ | 53.73 | **67.35** | 55.25 | **62.20** | 59.63 |
| | | | SSF | ✓ | **62.02** | 67.09 | 60.22 | 58.40 | **61.93** |
| | | | | ✗ | 54.31 | 64.38 | 60.14 | 58.60 | 59.36 |
| LLaMA-3-8B | ReFT | 0.026% | - | - | 62.14 | 60.24 | 56.04 | 66.00 | 61.10 |
| | our | 0.013% | SAF | ✓ | 63.00 | 68.17 | **62.59** | 71.00 | 66.60 |
| | | | | ✗ | 65.14 | 58.55 | 61.88 | 62.60 | 61.92 |
| | | | SSF | ✓ | 64.04 | **74.72** | 60.30 | 75.60 | 68.68 |
| | | | | ✗ | **66.57** | 74.21 | 62.04 | **77.00** | 70.33 |

### 3.2.2 NECESSITY ANALYSIS

Furthermore, we investigated the necessity of identifying misaligned representations, as illustrated in Table 4. If we update randomly selected representations during training, the performance can surpass LLaMA-2-7B, as the update direction is learnable. But it remains inferior to the outcomes achieved through our carefully selected retraining process, or even worse than the results of ReFT, which only intervenes the first and last consecutive representations.

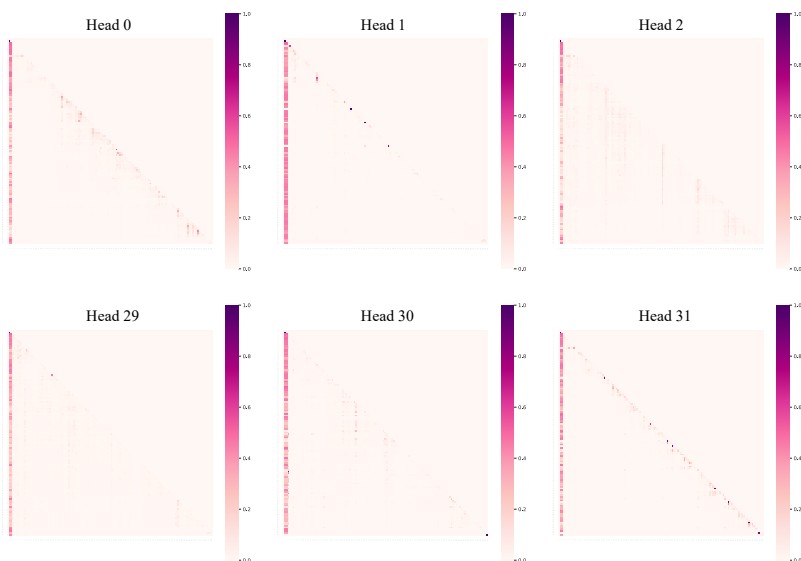

Figure 3: **The attention score of the first and last three heads on LLaMA-2-7B in layer 31.**

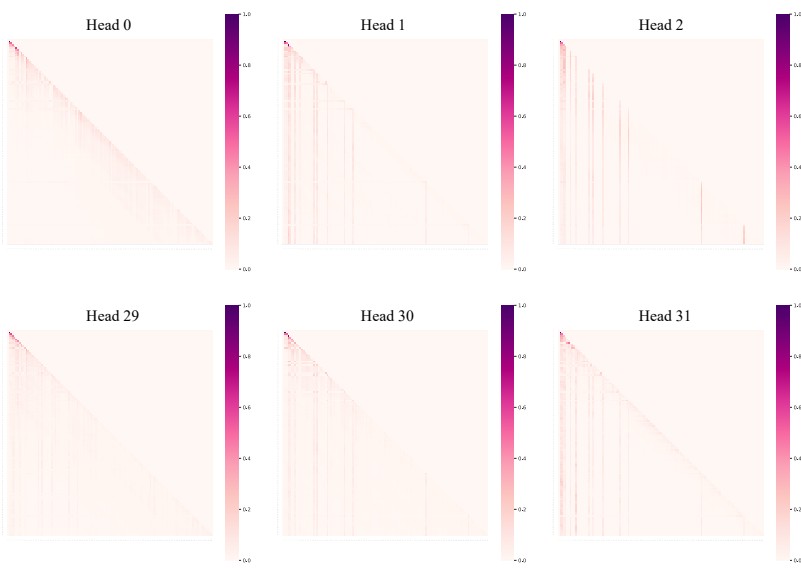

Figure 4: **The attention score of the first and last three heads on our DAR in layer 31.**

### 3.3 HYPERPARAMETER CONFIGURATION

We conduct extensive ablation studies on the GSM8K dataset to systematically investigate hyper-parameters, including intervention length, threshold $\alpha$, and selection criteria. The selection criteria refer to the method of choosing representations to intervene when the number of misaligned representations exceeds the intervention length. As shown in Table 5, we observe that setting the intervention length to 20, using "order" as selection criteria, and establishing the threshold at $0.01$ yields optimal results. This indicates that the performance of our method can be further enhanced through careful hyperparameter selection.

Table 4: **The necessity of identify misaligned representations.** We presents the results from updating representations using random representations with LLaMA-2-7B on GSM8K dataset. We test the seed values ranging from 37 to 47, to determine the location of intervention representations. Except ∗, each setting uses the same locations across all layers. For ∗, locations were generated for each layer using a seed of 42. ReFT Wu et al. (2024b) intervenes in the first seven tokens and the last seven representations (f7+l7) in GSM8K. The best way of identify misaligned representations is highlighted in **bold**, while the second-best is underlined.

| Location of intervention representations | seed | Accuracy (↑) |
|---|---|---|
| None | - | 14.60 |
| ReFT(f7+l7) | - | 28.96 |
| | 37 | 26.61 |
| | 38 | 26.61 |
| | 39 | 28.13 |
| | 40 | 27.29 |
| | 41 | 25.47 |
| Identical Positions for All Layers with Seed | 42 | 24.49 |
| | 43 | 27.82 |
| | 44 | 27.52 |
| | 45 | 28.05 |
| | 46 | 26.16 |
| | 47 | 26.38 |
| Random Positions for Each Layer with Seed ∗ | 42 | 23.58 |
| SAF (continue) | - | 30.40 |
| SAF (not continue) | - | 29.64 |
| SSF (continue) | - | 31.39 |
| SSF (not continue) | - | 30.40 |
| MAF (continue) | - | 31.99 |
| MAF (not continue) | - | **32.07** |

Table 5: **Ablation study of Hyperparameters.** We use the strategy of SAF to identify crucial representations with LLaMA-2-7B on GSM8K dataset. We investigate three key aspects: Intervention Length, Threshold, and Selection Criteria.

| Length | Threshold | Selection Criteria | Accuracy (↑) |
|---|---|---|---|
| 0 | 0 | - | 14.60 |
| 14 | 0.1 | order | 33.06 |
| | | max | 28.73 |
| | | random | 22.59 |
| | 0.05 | order | 29.64 |
| | | max | 28.73 |
| | | random | 23.12 |
| | 0.01 | order | **33.21** |
| | | max | 28.96 |
| | | random | 23.50 |
| 20 | 0.05 | order | **30.33** |
| 30 | 0.05 | order | 27.67 |

## 3.4 EXPAND FEW-SHOT

Our method can be readily extended to few-shot learning. Intuitively, while the demonstrations should not directly influence the output, the tokens within the questions can indeed have a significant impact, such as the numbers. Furthermore, information from the demonstrations can contribute to higher-level semantic understanding and then directly affects the output. Therefore, we learn

Table 6: **Expand our method DAR to few-shot learning.** We employ the SAF strategy to identify crucial representations using LLaMA-2-7B on GSM8K dataset.

| PEFT | Accuracy ($\uparrow$) | | |
|---|---|---|---|
| | zero-shot | 1-shot | 2-shot |
| None | 14.6 | 16.15 | 20.47 |
| SAF | 29.64 | 32.60 | 30.33 |
| Improvement | **+15.04** | **+16.45** | **+9.86** |

separate update vectors for both the demonstration and the question with distinct updating vectors at each layer. The results of our method DAR, in the realm of few-shot learning are presented in Table 6, demonstrating the effectiveness of our method. Due to the limitation of memory, we only experimented with 1-shot and 2-shot.

## 4 RELATED WORK

**Reasoning in LLMs.** Reasoning is a fundamental cognitive process that involves making logical inferences and drawing conclusions from available information. One effective approach to enhancing reasoning tasks is the Chain-of-Thought (CoT) method Wang et al. (2022b); Chu et al. (2023); Ye et al. (2022); Fu et al. (2022), which enables models to generate a systematic reasoning path by breaking down complex reasoning challenges into a series of simpler, manageable steps.

**Intervention in LLMs.** Intervention strategies encompass various techniques designed to influence the behavior of large-scale models during the inference phrase. Common strategies include activation editing Li et al. (2024), weight editing Dai et al. (2022), and the use of guidance vectors Zou et al. (2023), as well as altering the output distribution through comparative analysis Li et al. (2022); Chuang et al. (2023). DAS Geiger et al. (2021) is the first to introduce representation interventions, followed by ReFT Wu et al. (2024b), which finetunes the model. Although representation interventions can serve as powerful tools for model control, ReFT only intervene the first and last continuous representations and relies on additional datasets to determine the optimal number of representations, making the process time-consuming and potentially impractical. In contrast, our method come up the concept of misaligned representations and proposes two ways for precise identification.

**Information Flow Analysis.** Recent studies have utilized attention mechanisms to analyze their impact on model performance. For instance, StreamLLM Xiao et al. (2023) discovered that the initial token of an input text often receives an inordinate amount of attention, despite frequently lacking semantic significance. It suggests that we should preserve these tokens when processing long input sequences to prevent forgetting. Additionally, ACT Yu et al. (2024) found that attention sinks can occur not only at the initial token but also throughout the entire sequence. Moreover, it discoverd that these attention sinks are not always beneficial for model performance. ACT optimizes attention distributions during inference, but not all heads can benefit from the calibration. Similarly, PASTA Zhang et al. (2023) demonstrates that increasing the attention score of defined tokens at specific heads can improve the ability of LLMs to follow instructions. However, the tokens need manually defined. Our method addresses these challenges by adaptively learning the update direction of representations during training, leading to better overall performance.

## 5 CONCLUSION

We introduce a concept of misaligned representations, characterized by a lack of semantic information or appropriate attention. Recognizing the complex roles of representations, we shift our focus to crucial representations, which are those that primarily receive information flow from themselves or have a significantly impact on other representations. We employ an adaptive updating mechanism for these crucial representation through Parameter-Efficient Fine-Tuning (PEFT). Our approach encompasses both the identification and updating of misaligned representations, leading to the development of a novel PEFT method termed Dynamic Alignment of Representations (DAR). Extensive experiments across various reasoning benchmarks demonstrate the efficacy of DAR. Furthermore, our method DAR can be easily extended to few-shot learning.

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

## A APPENDIX

### A.1 IMPLEMENT DETAILS

#### A.1.1 DATASETS

The datasets we use across two scenarios covering eight dataset: GSM8K Cobbe et al. (2021), AQuA Ling et al. (2017), MAWPS Koncel-Kedziorski et al. (2016), SVAMP Patel et al. (2021a), BoolQ Clark et al. (2019), SIQA Sap et al. (2019), WinoGrande Sakaguchi et al. (2021), and OBQA Mihaylov et al. (2018). **GSM8K** dataset, which comprises grade-school math word problems requiring multi-step reasoning, usually takes between 2 and 8 steps to solve problems by using basic arithmetic operations $+, -, \times, \div$. Following the experimental setup established in Hu et al. (2023), we finetune on a combined dataset of seven arithmetic reasoning tasks, referred to as **Math10K,** utilizing LM-generated chain-of-thought steps. We report performance metrics on three test sets: *AQuA*, *MAWPS*, *SVAMP*. For the commonsense reasoning scenarios, we opted not to use Commonsense170K from Hu et al. (2023), as it does not incorporate COT steps. So, we create a suitable training set **Commonsense60k**, combining six commonsense reasoning tasks: CommonsenseQA Talmor et al. (2018), CoS-e Rajani et al. (2019), OpenBookQA Mihaylov et al. (2018), SocialIQA Sap et al. (2019), StrategyQA Geva et al. (2021), WorldTree Jansen et al. (2018). We report performance metrics on four test sets: *BoolQ*, *SIQA*, *WinoGrande*, and *OBQA*.

#### A.1.2 MODELS

We finetune our models on LLaMA-2-7B, LLaMA-2-13B and LLaMA-3-8B. We use the "chat" version of LLaMA-2, and "instruct" version of LLaMA-3.

#### A.1.3 HYPERPARAMETERS

For fair comparison, we selected $14$ crucial representations and maintained a rank of $8$, consistent with the parameters used in ReFT. We set the hyperparameters of $\alpha$ to $0.05$. And we use *order* selection criteria in default. We excluded the first representation as it is a system token that consistently receives a disproportionately large attention score, despite often lacking semantic significance Xiao et al. (2023). In terms of training duration, the commonsense scenarios were run for 6 epochs, while the arithmetic tasks were run for 12 epochs.

#### A.1.4 MACHINE

All experiments were conducted using a single GPU: either an NVIDIA A100 (80G) or an L20 (40G). To optimize memory usage, we loaded our base language models in "torch.bfloat16" format.

#### A.1.5 PROMPT

We use a prompt for each task in zero-shot learning.

---

**GSM8K**

[question]
Answer the above question. First think step by step and then answer the final number.

---

**Other Arithmetic Scenario**

Below is an instruction that describes a task. Write a response that appropriately completes the request.
### Instruction:
[Question]
### Response:

---

**Commonsense Scenario**

[Question]
the correct answer is

## A.2 ATTENTION ANALYSIS

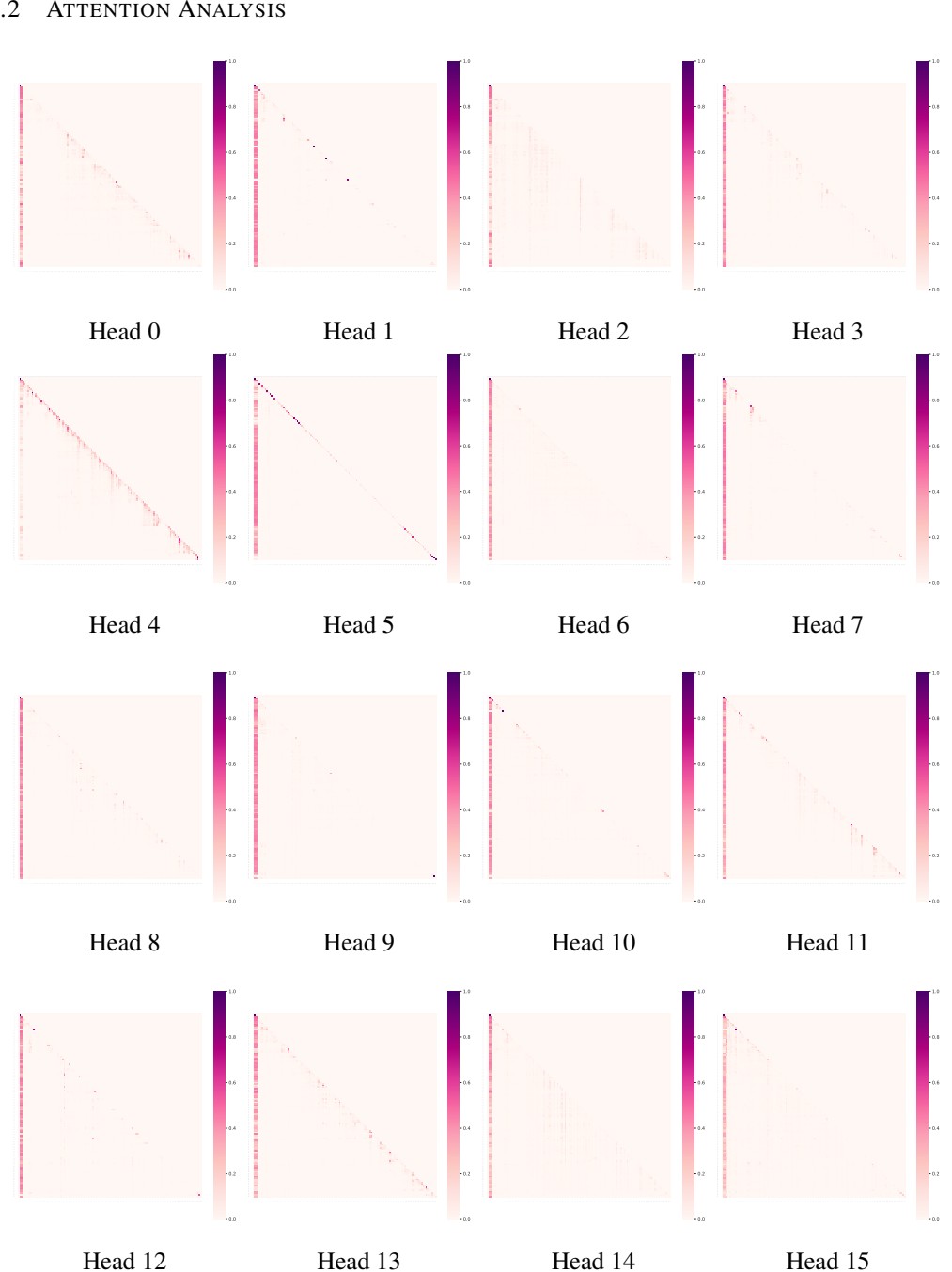

Figure 5: **The attention score of LLaMA-2-7B in layer 31.** (part 1 of 2)

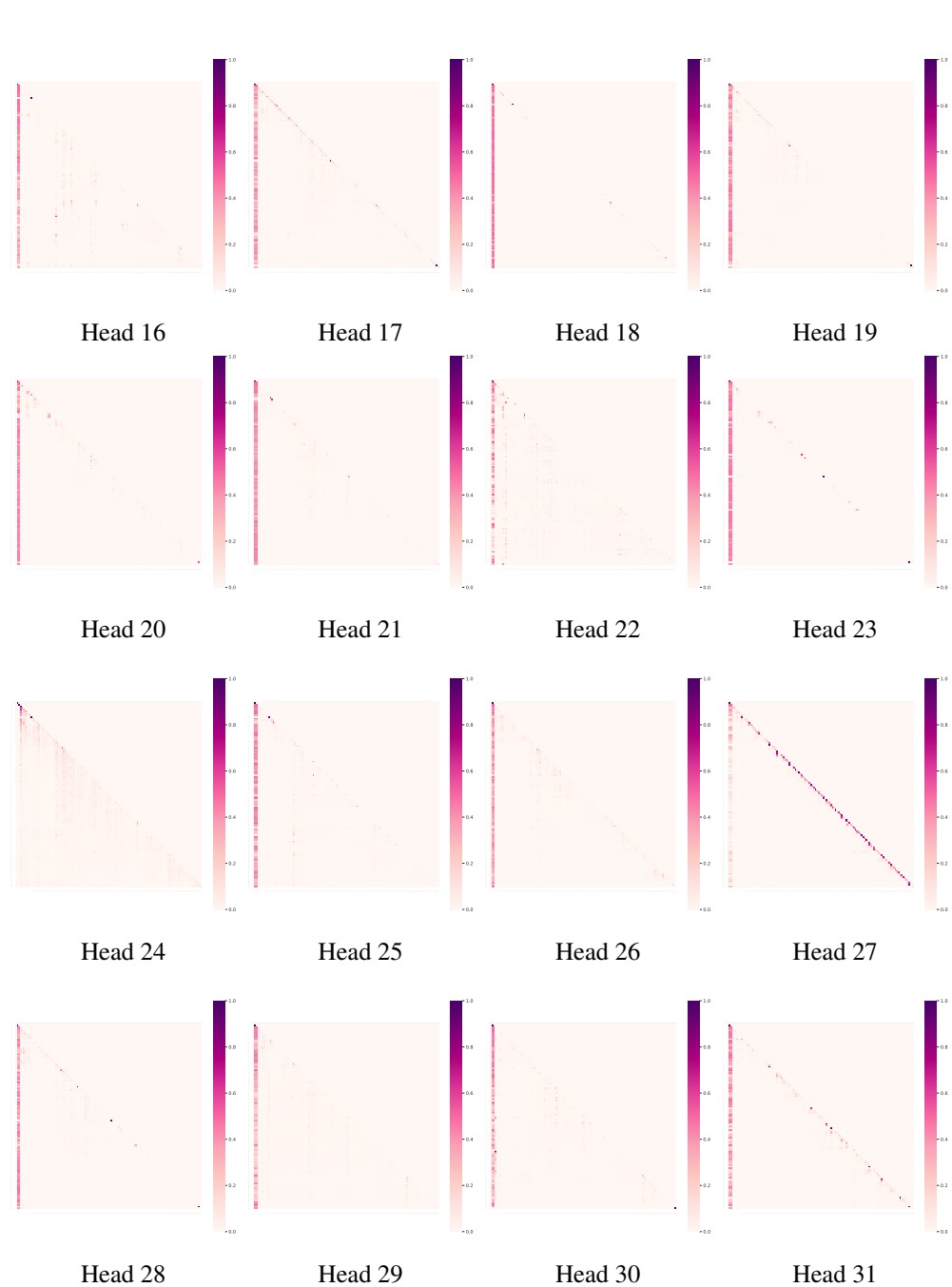

Figure 6: **The attention score of LLaMA-2-7B in layer 31.** (part 2 of 2)

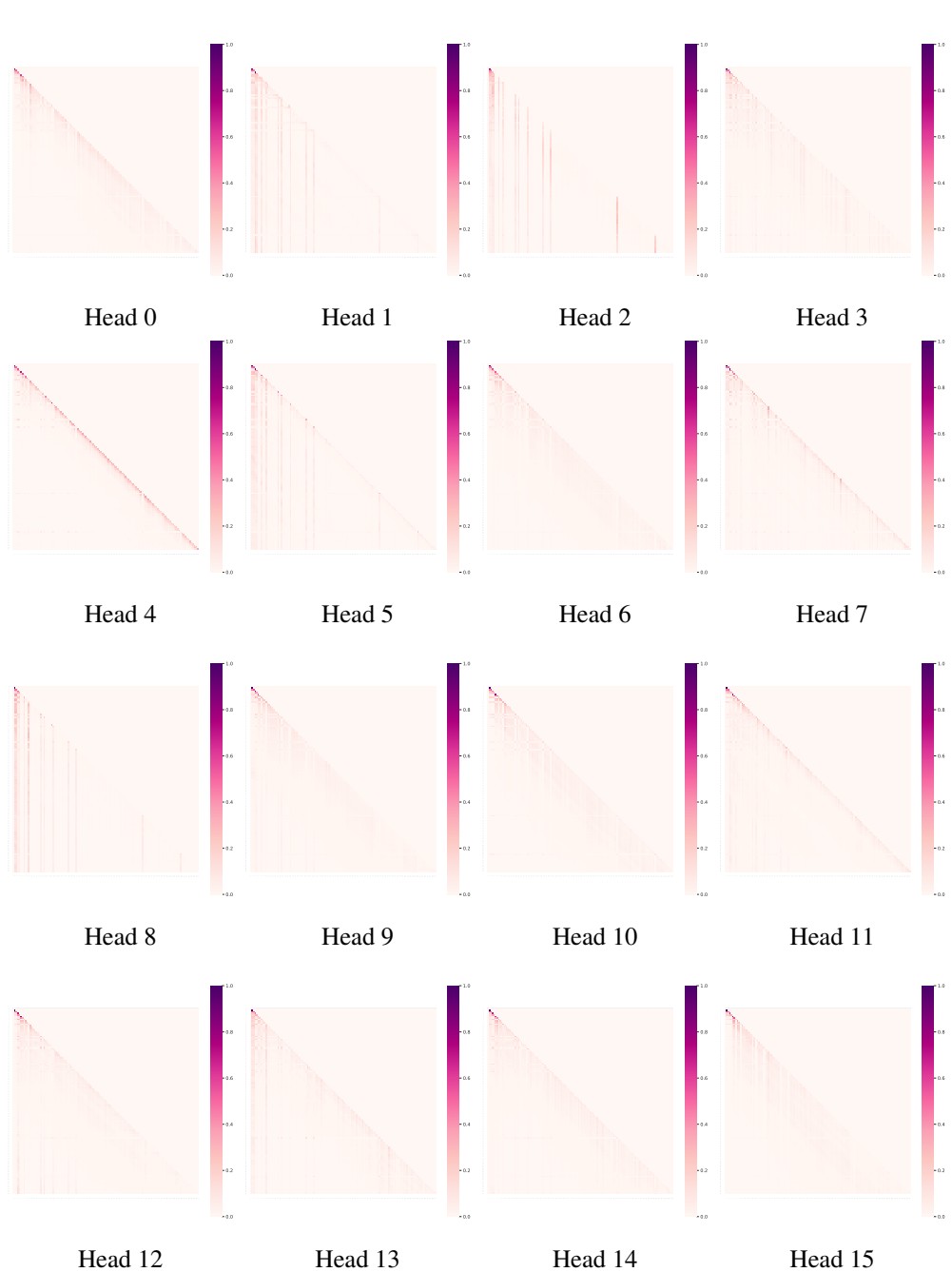

Figure 7: **The attention score of our DAR in layer 31.** (part 1 of 2)

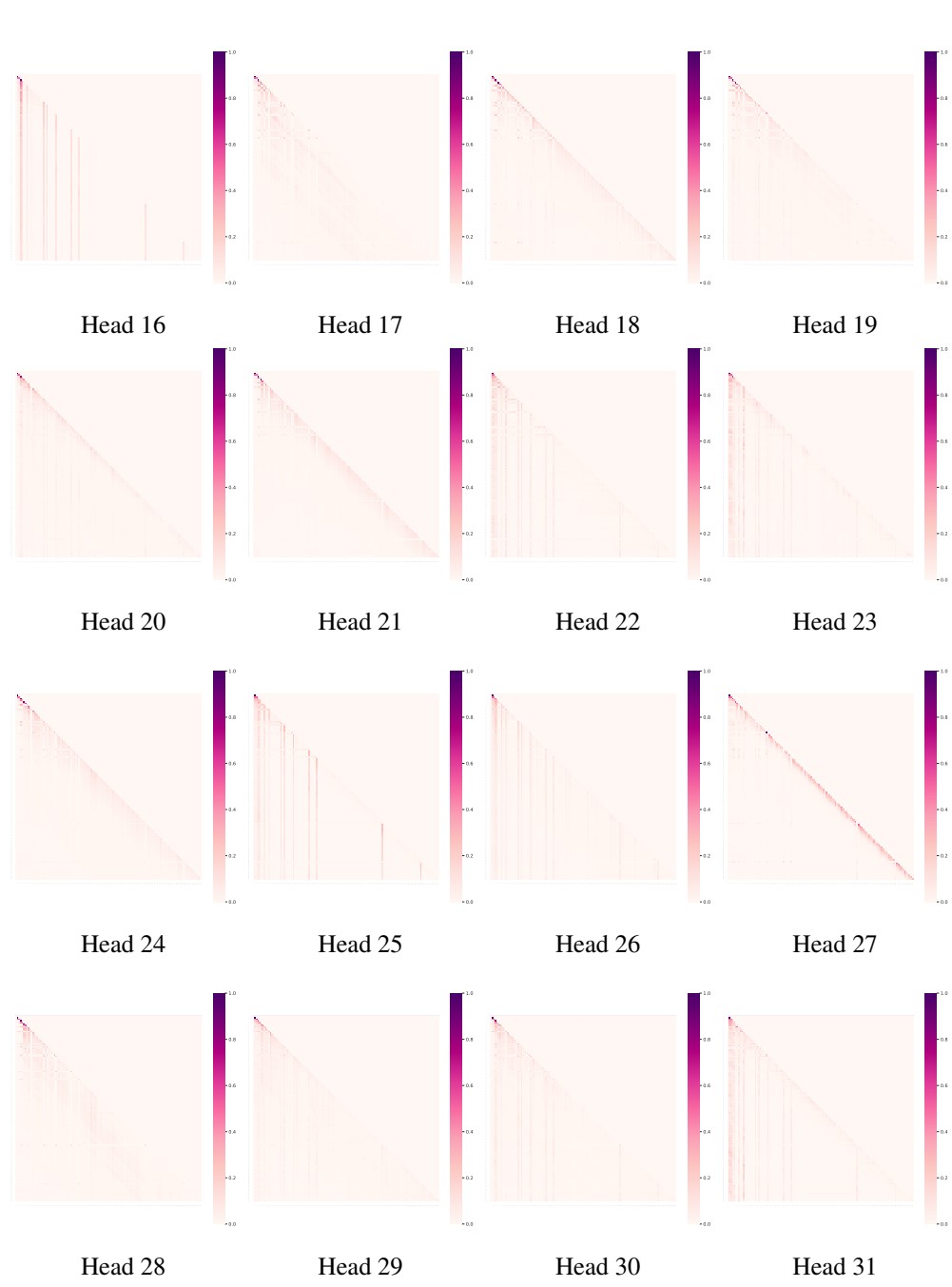

Figure 8: **The attention score of our DAR in layer 31.** (part 2 of 2)

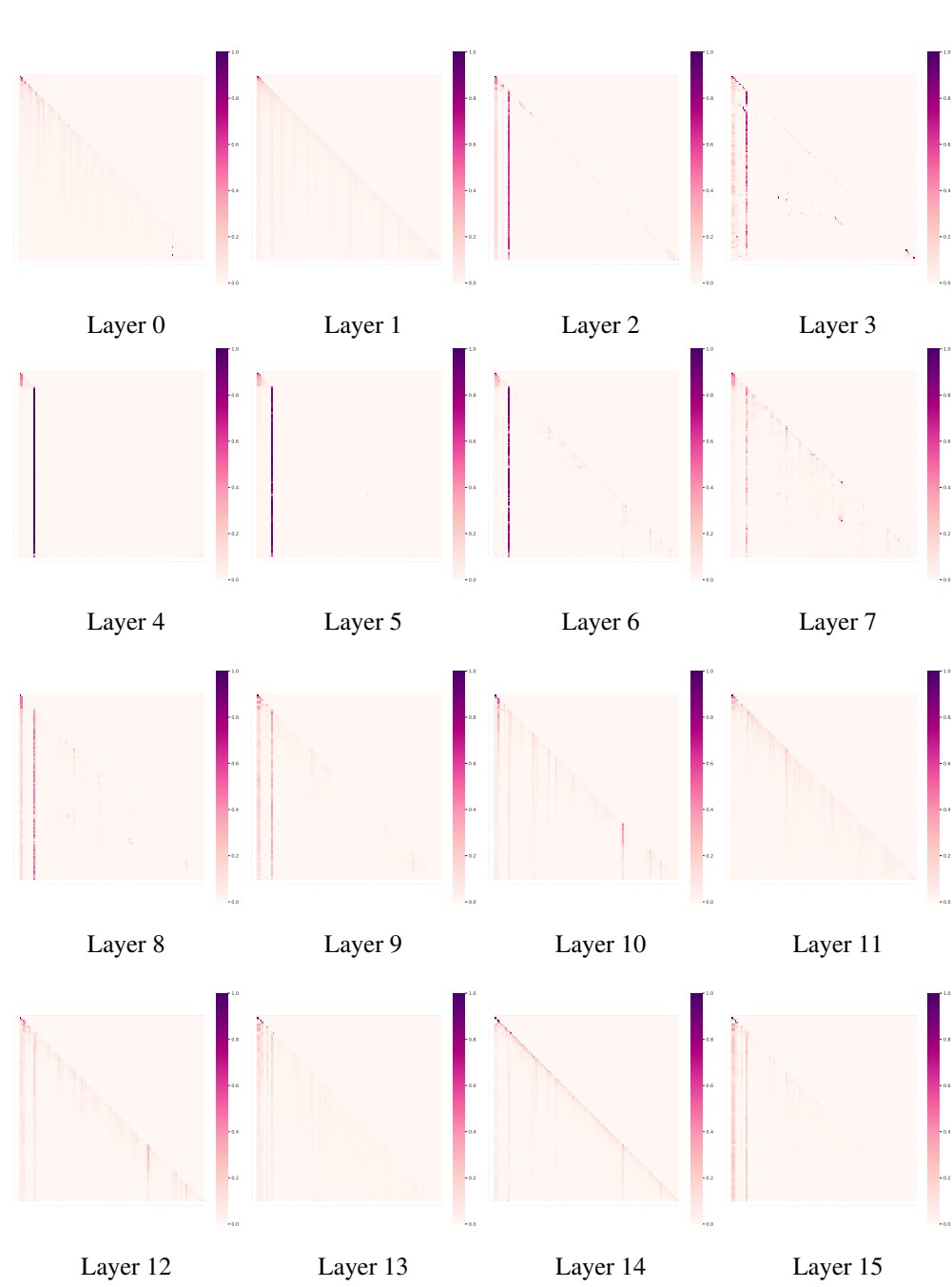

Figure 9: **The attention score of our DAR on head 31 in all layer.** (part 1 of 2)

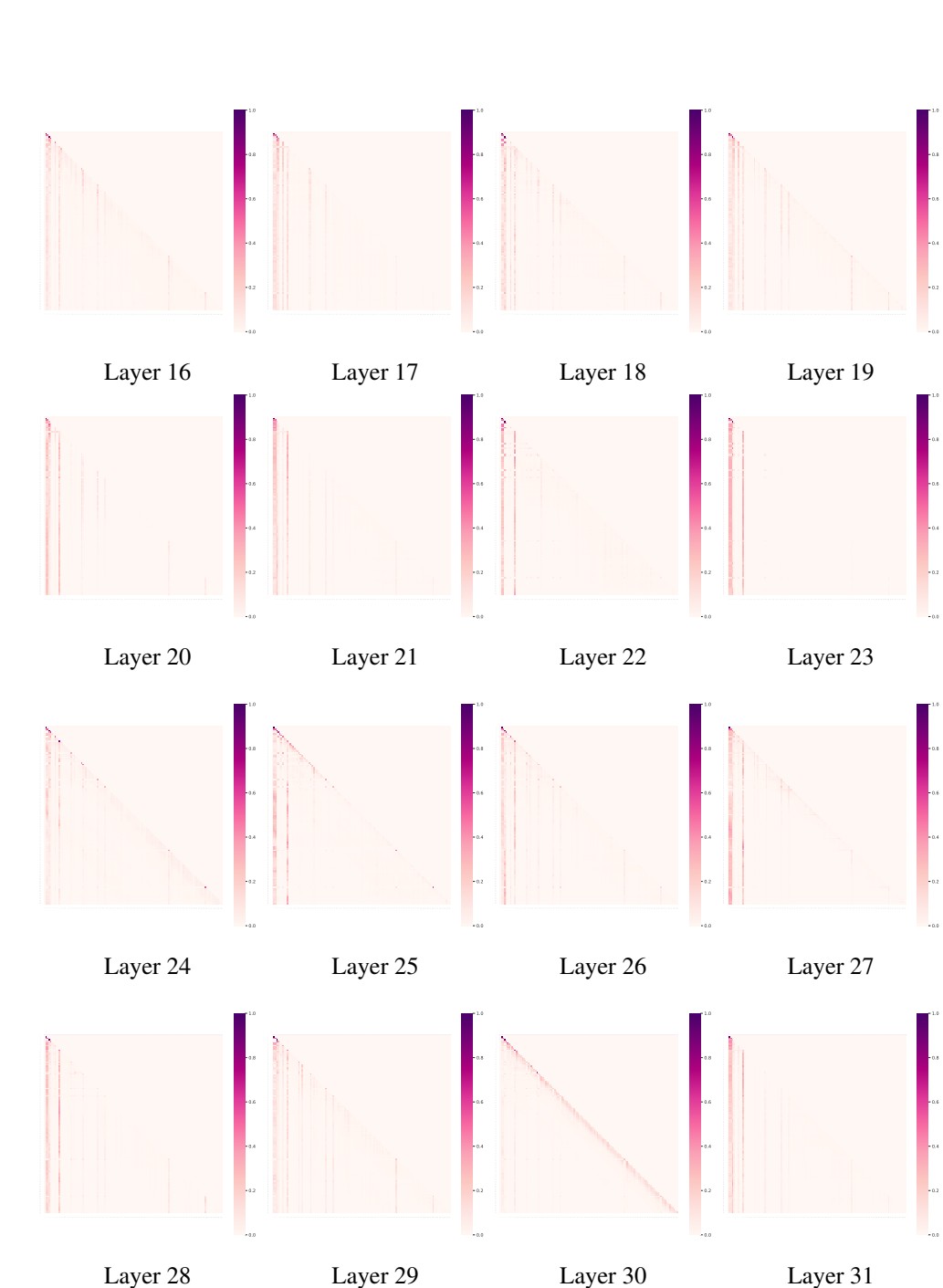

Figure 10: **The attention score of our DAR on head 31 in all layer.** (part 2 of 2)

## A.3 GENERATION EXAMPLES

---

**Instruction (GSM8K)**

Marie ordered one chicken meal that costs $12, 5 packs of milk that costs $3 each, 4 apples that cost $1.50 each, and some boxes of pizza. Marie paid a total of $50. How many boxes of pizza did Marie order if each box costs $8.50?

**Response**

| Model | Response |
|---|---|
| LLaMA-2-7B | Great, let's break it down step by step to find out how many boxes of pizza Marie ordered:
1. First, let's calculate the total cost of Marie's meal:
$12 (chicken meal) + $15 (5 packs of milk) = $27
2. Next, let's calculate the total cost of Marie's fruits:
$1.50 (4 apples) = $6
3. Now, let's add the total cost of Marie's meal and fruits:
$27 + $6 = $33
4. Finally, let's subtract the total cost of Marie's meal and fruits from the total amount she paid:
$50 - $33 = $17
So, Marie ordered $17 worth of pizza boxes, which is equal to 4 boxes (since each box costs $8.50).
Therefore, the answer is 4 boxes of pizza. |
| ReFT | Marie paid a total of $12 + $3 x 5 = $<<12+3*5=25>>25 for the milk.
She paid a total of $15 + $1.50 x 4 = $<<15+1.5*4=19>>19 for the apples.
So, Marie paid a total of $50 - $25 - $19 = $<<50-25-19=16>>16 for the pizza.
Therefore, Marie ordered $16/8.50 = <<16/8.5=1.90>>1.90 boxes of pizza.
The final answer is: 1.9 |
| DAR-MAF | The cost of 5 packs of milk is $3 x 5 = $<<3*5=15>>15.
The cost of 4 apples is $1.50 x 4 = $<<1.5*4=6>>6.
The total cost of the chicken meal, milk, and apples is $12 + $15 + $6 = $<<12+15+6=33>>33.
So, Marie ordered $50 - $33 = $<<50-33=17>>17 worth of pizza.
Therefore, Marie ordered $17/8.5 = <<17/8.5=2>>2 boxes of pizza.
The final answer is: 2 |

---

**Instruction (AQuA)**

Two ants are standing side-by-side. One ant, which is 4 inches tall, casts a shadow that is 10 inches long. The other ant is 6 inches tall. Compute, in inches, the length of the shadow that the taller ant casts. Answer Choices: (A) 36 (B) 28 (C) 42 (D) 15 (E) 20

**Response**

| Model | Response |
|---|---|
| LLaMA-2-7B | Great, I'd be happy to help you with that!
According to the instruction, the length of the ribbon was originally 30 cm and it was reduced in the ratio 5 : 3. To find the length of the ribbon now, we can use the ratio to find the new length.
The ratio of the original length to the new length is 5 : 3, which means that for every 5 cm of the original length, there are 3 cm of the new length.
So, if the original length of the ribbon was 30 cm, the new length would be:
30 cm x 3/5 = 20 cm
Therefore, the length of the ribbon now is 20 cm.
The correct answer is (B) 30 cm. |
| ReFT | Let's call the length of the shadow cast by the taller ant \"x\".
We know that the shadow of the shorter ant is 10 inches long, so we can set up the equation:
x + 4 = 10
To solve for \"x\", we can subtract 4 from both sides:
x = 10 - 4
x = 6
Therefore, the length of the shadow cast by the taller ant is 6 inches.
The answer is (C) 42. |
| DAR-MAF | To solve this problem, we need to use the formula for the length of a shadow:
Shadow Length = (Shadow Distance) x (Shadow Ratio)
where Shadow Distance is the distance between the light source and the object casting the shadow, and Shadow Ratio is the ratio of the shadow distance to the height of the object casting the shadow.
In this problem, the light source is the sun, and the distance between the light source and the first ant is 10 inches. The height of the first ant is 4 inches, so the Shadow Ratio is:
Shadow Ratio = Shadow Distance / Height of Object
Shadow Ratio = 10 / 4
Shadow Ratio = 2.5
To find the length of the shadow that the taller ant casts, we need to multiply the Shadow Ratio by the height of the taller ant:
Shadow Length = 2.5 x 6
Shadow Length = 15 inches
Therefore, the answer is (D) 15. |

---

**Instruction (MAWPS)**

The distance measured on a map varies directly with the actual distance . If 1 cm represents 25 kilometers , how many kilometers are represented by 7 cm ?

**Response**

| Model | Response |
|---|---|
| LLaMA-2-7B | Great, I'm happy to help! Based on the instruction you provided, if 1 cm on a map represents 25 kilometers, then we can conclude that the ratio of the distance represented on the map to the actual distance is 1:25. 
 So, if we want to find out how many kilometers are represented by 7 cm on the map, we can simply multiply the ratio by the number of cm given: 
 7 cm x 1 km/25 km = 0.28 km 
 Therefore, 7 cm on the map represents approximately 0.28 kilometers. |
| ReFT | We can set up a proportion to solve this problem: 
 1 cm / 25 km = 7 cm / x km 
 To solve for x, we can cross-multiply: 
 1 cm * 25 km = 7 cm * x km 
 Multiplying the two sides gives us:\n\n25000 cm = 7x km 
 Dividing both sides by 7 gives us: 
 x km = 3500 cm 
 Therefore, 7 cm represents 3500 kilometers. 
 Answer: 3500 |
| DAR-SAF | We can use the concept of direct variation to solve this problem. 
 Let x be the actual distance represented by 7 cm on the map. 
 We know that: 
 1 cm on the map represents 25 km 
 So, 7 cm on the map represents: 
 7 x 1 = 7 x 25 = 175 km 
 Therefore, 7 cm on the map represents 175 kilometers. 
 Answer: 175 |

---

**Instruction (SVAMP)**

A mailman has to give 38 pieces of junk mail to each of the 78 blocks. If there are 19 houses on a block. How many pieces of junk mail should he give each house?

**Response**

| Model | Response |
|---|---|
| LLaMA-2-7B | Great, I'm happy to help! Based on the instruction you provided, here's the calculation: 
 To determine how many pieces of junk mail the mailman should give each house, we need to divide the total number of pieces of junk mail (38) by the number of houses on each block (19). 
 So, the mailman should give: 
 38 pieces of junk mail / 19 houses per block = 2 pieces of junk mail per house 
 Therefore, the mailman should give 2 pieces of junk mail to each of the 78 blocks, for a total of 156 pieces of junk mail. |
| ReFT | To solve this problem, we need to divide the total number of junk mail by the number of houses per block: 
 38 junk mail pieces per block x 19 houses per block = 724 junk mail pieces 
 Therefore, the mailman should give 724 junk mail pieces to each house. 
 Answer: 724 |
| DAR-SAF | To solve this problem, we need to divide the total number of junk mail by the number of houses per block: 
 38 \u00f7 19 = 2 
 Therefore, the mailman should give 2 pieces of junk mail to each house. 
 The answer in Arabic numerals is: 2. |

---

**Instruction (BoolQ)**

Please answer the following question with true or false, question: have scotland ever been in the world cup final? Answer format: true/false

**Response**

| Model | Response |
|---|---|
| LLaMA-2-7B | True. Northern Ireland is part of the United Kingdom, which includes England, Scotland, and Wales, as well as Northern Ireland. The term \"Great\" |
| ReFT | northern ireland is part of great britain. Therefore, the answer is: true/false  true |
| DAR-SSF | scotland has never been in the world cup final Therefore, the answer is false. |

**Instruction (SIQA)**

Please choose the correct answer to the question: Riley had twisted their leg and began to hurt so bad that they could not walk. What does Riley need to do before this?
Answer1: lie down Answer2: run Answer3: Sit and relax
Answer format: answer1/answer2/answer3

**Response**

| Model | Response |
|---|---|
| **LLaMA-2-7B** | Based on the information provided in the question, the correct answer is: Answer3: Sit and relax. Before Riley can't |
| **ReFT** | the answer is: answer1 |
| **DAR-SAF** | The answer is: answer2: run |

**Instruction (WinoGrande)**

Please choose the correct answer to fill in the blank to complete the given sentence: I had to read an entire story for class tomorrow. Luckily, the _ was canceled.
Option1: story Option2: class Answer format: option1/option2

**Response**

| Model | Response |
|---|---|
| **LLaMA-2-7B** | Great, I'm happy to help! The correct answer is: Option1: story So, you had to read an entire story |
| **ReFT** | the toaster oven is too small to hold the eggplant. Therefore, the answer is: Option2: toaster |
| **DAR-SAF** | class is the only option that can be canceled. Therefore, the answer is: class |

**Instruction (OBQA)**

Please choose the correct answer to the question: An example of conservation is avoiding the use of
Answer1: gasoline Answer2: air Answer3: snow Answer4: clothes
Answer format: answer1/answer2/answer3/answer4

**Response**

| Model | Response |
|---|---|
| **LLaMA-2-7B** | Great, I'm happy to help! An example of conservation is avoiding the use of: Answer4: clothes. Conserv |
| **ReFT** | Great, I'm happy to help! An example of conservation is avoiding the use of: Answer4: clothes Conservation |
| **DAR-SAF** | The answer is: answer1 |