# OpenReview forum: "Dynamic Alignment of Representations for Enhanced Chain-of-Thought Reasoning in Large Language Models"
_ICLR.cc/2025/Conference — ICLR 2025 Conference Withdrawn Submission_

### Official Review · Reviewer_bZx5 · 2024-10-28

**Soundness:** 3
**Presentation:** 2
**Contribution:** 3
**Rating:** 5
**Confidence:** 4

**Summary:**

- This study proposes an efficient representational fine-tuning method which dynamically selects and intervene representations at the instance level.
- The experiment results validate the effectiveness of the proposed method on arithmetic and commonsense on three base models: LLaMA-2-7B, LLaMA-2-13B, and LLaMA-3-8B.
- This method is proven to be extended to few-shot learning.

**Strengths:**

- This study proposed an efficient dynamic representational fine-tuning method. The proposed method is novel.
- The authors conducted a wide range of experiments to validate the proposed method with eight diverse datasets across two scenarios, arithmetic and commonsense, and three base models.

**Weaknesses:**

- The proposed method employs the same modular addition method as the previous study, ReFT, with the only difference being the dynamic finding of important representations via filtering. This may limit the impact of the novelty.
- Regarding the experimental results (Table 1-3), the proposed method has dramatically fewer trainable parameters than LoRA, but the difference is smaller than ReFT, which is the basis of the proposed method, by about 1/2. In my understanding, the difference is attributed to the hyper-parameter setting, such as the number of intervening tokens or dimension size of low-rank projection matrix. In addition, the proposed method requires running the models once to find the misaligned representation and then again to edit the representation, which doubles the total computational effort required. So, the total computational cost of the proposed method may nearly equal to ReFT.
- The study did not discuss which filtering is the better as a conclusion among SAF, MAF, and MSF.
- Lack of explanations in 3.Experiment section. For example, the explanation of the word "continue" in Table 1-3 is not present in the main text. There is a few explanation in the caption of Table 1: "The ✓ means that the misaligned representations is identified from the misaligned representations in the previous layer." but this is not sufficient explanation for readers to understandable. Overall, much information is not included in the main text but with a richer description in the table captions.

**Questions:**

- The authors propose the following three filtering approach: Self-Referential Attention Filtering (SAF), Self-Referential Saliency Filtering (SSF), and Multi-Referential Attention Filtering (MAF). As a natural conception, why not try MSF (Multi-Referential Saliency Filtering)?
- I don't think that the representation found via the proposed filtering is necessarily a misaligned representation because the filtering process just finds out the impactful tokens through the lens of attention. It may be a terminological issue of how to define the word "misalignment", but what do authors think about this?
- Equation(2): Is it a standard attention score of Transformer (but there is no symbols like key and query), or is it a newly proposed attention score for this study? (Moreover, h^l_l seems to be weired.)

---

### Official Review · Reviewer_bXCN · 2024-10-30

**Soundness:** 2
**Presentation:** 3
**Contribution:** 3
**Rating:** 5
**Confidence:** 2

**Summary:**

This paper introduces Dynamic Alignment of Representations (DAR), a novel parameter-efficient fine-tuning method for improving chain-of-thought reasoning in large language models. The method identifies and updates "crucial representations" that are likely to contain problematic misaligned representations, using two key strategies: self-referential filtering (representations receiving information mainly from themselves) and multi-referential filtering (representations influencing multiple others). Evaluated across eight datasets using LLaMA models, DAR achieves significant improvements over baseline methods while using far fewer parameters - achieving 17.47% improvement over base LLaMA-2-7B on GSM8K while using 51x fewer parameters than LoRA. The method proves effective for both arithmetic and commonsense reasoning tasks and extends well to few-shot learning scenarios.

**Strengths:**

1.  Instead of using fixed intervention points like previous work, this paper introduce a new concept of "misaligned representations" and develop a method to identify them through crucial representations. Their dual-path method combining attention and saliency scores feels like a natural but novel extension of existing ideas.

2. The author validated their method across 8 different datasets using 3 different model sizes. The results are quite remarkable - achieving a 17.47% improvement on GSM8K while using 51x fewer parameters than LoRA shows both the effectiveness and efficiency of their method. The ablation studies are comprehensive and help build confidence in the approach.

3. The paper is also well written. The methodology was explained step by step, with clear diagrams and intuitive examples supporting the mathematical formulations.

**Weaknesses:**

1. The relationship between self-referential and multi-referential filtering is not deeply explored. It would be valuable to understand when one approach might be preferable over the other.

2. The base model comparison focuses only on the LLaMA family. Testing on other architectures (e.g., Mistral) would better demonstrate the method's generalizability.

3. While parameter efficiency is demonstrated, there's no discussion of training efficiency or convergence compared to other methods.

**Questions:**

1， Have you observed any patterns in which type of filtering (self-referential vs multi-referential) works better for different types of tasks?

2， The threshold α seems to significantly impact performance. Do you have any guidelines for selecting this parameter for new tasks?

---

### Official Review · Reviewer_PufB · 2024-11-04

**Soundness:** 1
**Presentation:** 1
**Contribution:** 1
**Rating:** 3
**Confidence:** 4

**Summary:**

This paper proposes to improve representation fine-tuning (ReFT), a parameter-efficient method for selecting and linearly projecting certain hidden states in an LM at inference time which can improve performance on certain tasks. The paper proposes a method for better finding *which* representations in a model to change: instead of treating this as a hyperparameter selected using a validation set, the authors propose to select representations based on those that have a significant influence on other representations and/or on the future representations at the same token position (where influence is measured using attention scores and saliency metrics). The authors demonstrate performance gains over both ReFT and randomly selecting representation positions on a variety of tasks for Llama models of various sizes and versions.

**Strengths:**

- Originality: to the best of my knowledge, there hasn't been work on changing how ReFT method selects representations to edit. The proposal to select important representations  on an instance-level basis as those with a large effect on later operations of the network makes sense.
- Significance: parameter-efficient finetuning is a valid and important direction for research. The authors' gains over ReFT seem consistently substantial on 3 different models and 8 datasets (though see below concerns about results).

**Weaknesses:**

- Contribution: the abstract & introduction fail to convey the problem that the paper is trying to solve. This is partially because a) the term "representations" seems to be overloaded to mean both textual input tokens and hidden states, b) necessary background on the ReFT method from prior work is not provided, and c) important/key terms that are proposed like "misaligned representations" and "crucial representations" aren't ever formally defined. I followed the problem laid out in Figure 1, but did not understand Figure 2 at all, in part because the notation in the Figure isn't explained.
- Contribution: the stated claim of parameter efficiency (51x fewer params than LORA) is not unique to the proposed method, since the prior work this work builds off of/compares to, ReFT, achieves 15-65x fewer parameters.
- Related Work: the related works section is lacking a lot of historical work on information flow and attention pattern analysis in NLP (just to name a couple, see https://aclanthology.org/2020.acl-main.385/, https://aclanthology.org/P19-1580/, etc.) as well as on intervention on LLM representations (see, e.g., the ReFT paper's related work for reference)
- Method: some key details of the proposed method aren't explained and/or are extremely difficult to follow. To name a few- how are the most important representations chosen, given the proposed metrics in Section 2.2? Is there a threshold? A hyperparameter for how many to select? It appears there is a method for this based on Table 5, but they're not described anywhere in the main text. Is the update rule proposed in Section 2.3 identical to that in prior work? A lot of the results seem to hinge on something called "Continue" which is never described.
- Results: it is not described how hyperparameter search is conducted for the ReFT baseline or the proposed methods, which raises doubt over whether the stated gains of the proposed method over ReFT may be due to more tuning. Also, why are the performances you report for ReFT so much lower than those reported in the original paper? This makes it difficult for me to have full confidence in the presented results.
- Writing: overall, I found the paper quite difficult to understand, missing key details, and in need of substantial writing improvements. I elaborated on some more writing issues in "Questions" below.

**Questions:**

Comments/suggestions:
- citations that are not explicitly the subject of a sentence should use `\citep{}` instead of `\citet{}`
- many typos & grammar errors throughout (found 11 on first 3 pages): would recommend a thorough proofread. This is particularly noticeable in Fig. 1
- experiments are conducted on only one model family (Llama)
- notation issues, e.g., $M(\mathbf{h})$ doesn't really make sense to indicate a subset of $\mathbf{h}$. Values $f$ and $l$ of ReFT are never defined. Many other undefined or not clearly defined variables in Section 2.2.
- why must misaligned representations be a subset of crucial representations?
- citations should be provided for saliency scores. Attention scores as a signal of influence should be considered after the dot product is taken; see https://aclanthology.org/2020.emnlp-main.574/ for more discussion

---

### Official Review · Reviewer_iyLj · 2024-11-04

**Soundness:** 3
**Presentation:** 2
**Contribution:** 2
**Rating:** 3
**Confidence:** 3

**Summary:**

This paper introduces Dynamic Alignment of Representations (DAR), a novel parameter-efficient fine-tuning method designed to enhance chain-of-thought reasoning in large language models. Unlike existing approaches that focus on general categories or continuous blocks of representations, DAR dynamically identifies and updates misaligned representations at the instance level. The method operates through two key mechanisms: identifying crucial representations via self-referential and multi-referential filtering and updating these representations through adaptive learning while keeping the base model frozen. The effectiveness of DAR was demonstrated across eight diverse datasets involving arithmetic and commonsense reasoning tasks, using LLaMA-2-7B, LLaMA-2-13B, and LLaMA-3-8B as base models. The method's efficiency and adaptability, including its potential for few-shot learning applications, make it a significant contribution to the field of language model fine-tuning.

**Strengths:**

1. This paper introduces a novel dynamic approach to identifying misaligned representations. Meanwhile, it also develops a unique dual-filtering mechanism (self-referential and multi-referential) and proposes an adaptive updating strategy for representations.
2. The method is very efficient and only uses 51 times fewer parameters compared to LoRA.
3. The authors also conducted a lot of experiments to show the effectiveness of their method across 8 diverse datasets, covering both arithmetic and commonsense reasoning tasks. Also, the experiments verify the methods on three different base models, including Llama2-7B, Llama2-13B, and Llama3-8B.

**Weaknesses:**

1. Some parts of the paper are hard to follow. For example, Section 2 refers to Figure 1 multiple times when discussing how to identify crucial tokens based on their representations. However, Figure 1 doesn't contain any representations at all. Also, there is no detailed explanation of Figure 1, even in its caption and the Introduction, making the example quite confusing. It seems that "per" is the only crucial token identified, and only changing "per" to "a" can lead to the correct answer.
2. The paper interchangeably used "misaligned representations" and "crucial representations." The authors propose two methods to identify crucial representation in Sections 2.1 and 2.2. Here, they detect the attention score and select the tokens with high incoming and outgoing attention scores. However, in all other parts of the paper (e.g., abstract, introduction, and other subsections in the method), the authors automatically treat those crucial representations as misaligned tokens without further explanation. This pose a significant doubt about the motivation of the paper.
3. The potential sensitivity to hyperparameter tuning isn't extensively analyzed. The paper doesn't thoroughly discuss the impact of the threshold in self-referential filtering.

**Questions:**

No

---

### Note · Authors · 2024-11-18

I have read and agree with the venue's withdrawal policy on behalf of myself and my co-authors.